# Mother-Tongue Biblical Hermeneutics and the Pursuit of Ethnic Harmony in Ghana

Frederick Mawusi Amevenku [1,2]

1   Institute of Mother Tongue Biblical Hermeneutics, Trinity Theological Seminary, Legon, Accra, Ghana;
    fm.amevenku@trinity.edu.gh
2   Department of Old and New Testament, Faculty of Theology, Stellebosh University, Stellenbosch 7602,
    South Africa

**Abstract:** Ethnic disharmony exists among the people of Ghana. What are the reasons for this? Ghana is an amalgamation of different ethnic groups, cultures, peoples and states to form one entity. The partition of Africa is mainly responsible for this, though there are other contributing factors. The project to partition Africa led, in large measure, to the erosion of the African identity. The 19th- and 20th-century European Christian mission to southern West Africa exploited this reality to their mission advantage. Unfortunately, the result seems to be counterproductive because the mission project, for the most part, produced a version of Christianity that failed to affirm the African identity. Concerned Africans, now on a mission to deconstruct the imperialist, European mission-constructed West African Christian identity, realise that biblical interpretation is one major source of this decolonial agenda. How does a mother-tongue reading of Ephesians 2 help decolonise Eʋe Christianity and promote ethnic harmony in Ghana? Using mother-tongue biblical hermeneutics, this paper argues that the pursuit of ethnic harmony in Ghana is a decolonial hermeneutic with potential for fostering ethnic harmony in Ghana.

**Keywords:** ethnic harmony; mother-tongue hermeneutics; partition of Africa; decolonial hermeneutic





## 1. Introduction

According to the Ghana Statistical Service, Ghana is a multi-ethnic, multicultural, multi-linguistic and multi-religious West African country. Ghana has more than eight major ethnic groups, including Akan (about 47%), Mole-Dagbani (about 18%), Eʋe (about 14%) and Ga-Dangbe (about 7%). Ghana is the second most populous West African country after Nigeria. The population of Ghana is estimated at 34 million people in 2023.

Ethnic diversity in Ghana is both an opportunity and a challenge. Positively, ethnic diversity can be harnessed to develop the country, since each ethnic group possesses varied skills and resources to contribute to improving Ghanaian society. Nevertheless, some Ghanaian people exploit ethnic diversity to fester irreconcilable divisions that tend to be a bane to society. How do these nuances of Ghanaian society play out? How might a mother-tongue hermeneutic study of Eph. 2:11 foster the pursuit of ethnic harmony in Ghana? This paper attempts a solution to the problem of ethnic disharmony in Ghana through a mother-tongue hermeneutic reading of Eph. 2:11.

## 2. An Overview of Ephesians

The Epistle to the Ephesians is attributed to Paul (v.1), but some scholars argue that it could not have been written by Paul. Those scholars give several reasons. The right of any scholar to express a view on a Bible text is affirmed. The position adopted in this paper, however, is simply to affirm what the canonised text says: that Paul is the author of the epistle. Stemming from this is the message of the epistle that we are interested in, particularly in Section 2, from where it has been observed the author addresses disharmony

between Jews and Gentiles, which Christ came to heal. God had long previously planned, according to Paul, to send Christ into the world to unite Jews and Gentiles as part of his exercise of divine sovereignty over *ta panta*, including humanity (1:10). The six chapters of the epistle can conveniently be divided into two: the first section outlines critical Christian beliefs, previously hidden, when God had not fully revealed his salvation plan to unite everything in heaven, on earth and the subterranean under Jesus, his son. Now that the plan has been revealed, believers in Christ are already seated with Christ in heavenly places. The most significant of these blessings is the salvation that Christ offers.

According to Section 2, humanity had been dead because of sin (2:1–3), but God, through Christ, saved humanity by his mercy and love (2:4–9). The result of this salvation is peace with God and reconciliation with the created order, among which is the unity of Jews and Gentiles into one humanity (2:11–22), resulting in complete ethnic harmony for those who belong to Christ.

God's action in Christ brought an end to ethnic labelling (2:11), disharmony and hostility, and the author urges his readers to remember this fact. The readers would remember that they were once 'aliens from the commonwealth of Israel and strangers to the covenants of promise, having no hope and without God in the world' (2:12). The blood of Christ bridged the gap and secured peace (2:13), ending all hostility between Jews and Gentiles (2:14). Christ abolished the law 'with its commandments and ordinances' (2:15a) and created a 'one new humanity in place of the two, thus making peace' (2:15b) by reconciling both to God as a single body (2:16). Through Christ, then, both Jews who were near and Gentiles who were far off have access to one Spirit of the Father (2:18). He has, therefore, made Jews and Gentiles 'fellow citizens with the saints and…members of the household of God' (2:19), 'built upon the foundation of the apostles and prophets, with Christ Jesus himself as the cornerstone' (2:20). Christ has, thus, built a 'holy temple in the Lord' (2:21), a spiritual dwelling place of God (2:22). The series of powerful metaphors employed in Verses 19–22 point to the new humanity reconciled and made one in Christ as a household of God, in God's *polis*, a building still under construction to promote further citizenship in God's new community.

Members of this new community are 'fellow citizens' (v. 19), whether Jew or Gentile, belonging to a 'city and province' (Kobelski 2000, p. 888) that 'transcends…political boundaries…and extends to fellowship with angels' (Kobelski 2000, p. 888). In other words, identity formation is at the heart of the Epistle to the Ephesians (Talbert 2007). 'The basic social unit of Greco-Roman society was the household, which included parents, children and slaves' (Kobelski 2000, p. 888). Similarly, we can judge from 2:20–21 that the basic social unit of the new community of God's children, the Christian household, includes God as parent, Christ the Son and cornerstone, apostles and prophets as foundations of God's building and all believers in Christ as children of the household. Of course, there are no slaves anymore because, unlike the Greco-Roman household of antiquity, the Christian household is a new, renewed and transformed community (see Gal. 3:28). Christ, therefore, destroyed, through reconciliation, the ethnic tension that previously characterised the relationship between the two humanities (Merida 2014). This saving work of Christ can be expressed in a 'cause and effect' fashion (Swindoll 2015, p. 225) because it is the effect of God's grace that brought near those who had previously been separated, excluded and estranged (2:12).

Having been raised and seated with Christ in the heavenly places, believers are alive in Christ (2:1–10), though before they had been alienated. After the period 'before', they are united with Christ and with the redeemed family of God. The rhetoric of 'before and after' (Eisenberg 2019) moves the teaching to universal application in 2:1–10 to the level of specific application to *ta ethnē* in 2:11–22. The peace offering (death of Jesus) that secured the reconciliation of the *ta ethnē* both with God and with the Jews (the circumcision) is a salvation blessing with horizontal and vertical dimensions (O'Brien 1999, p. 182), enabling *ta ethnē,* having been included in the one body of Christ (O'Brien 1999, p. 182), to recall their former plight and their plight now, to avoid relapse and a feeling of inferiority to the Jewish

race. Paul, in discussing the benefits of divine salvation in 2:11–12 for *ta ethnē*, secures a reminder of the believers' 'prior alienation from God' (Merida 2014, p. 42) and God's chosen people, which alienation Christ destroyed by means of reconciliation (Merida 2014). It has religious (Godward), cultural (rituals, feasts and sacrifice) and racial (Abrahamic ancestry) dimensions.

Verse 14 shows that the reconciliation that Christ effected through his death on the cross also includes peace and unity between Jews and Gentiles because, as the author says of Christ, 'he is our peace' (Kobelski 2000, p. 888). The barrier of hostility that Christ is said to have torn down may well point figuratively to the physical wall of partition between the Jewish and Gentile courts in the Jewish temple in Jerusalem as Josephus observes (Josephus *Antiquities, Book,* 15.11.5 n.d.), but the reference to 'enmity' 'suggests that the image is intended to depict the end of ethnic hostility between the two groups' (Kobelski 2000, p. 888).

### 3. An Eυe Mother-Tongue Hermeneutical Reading of Ephesians 2

John Ekem coined the term 'mother-tongue biblical hermeneutics' (MTBH) in 2009. The idea and concept of studying the Bible in one's mother tongue in Ghana did not originate with Ekem, however. Kwesi Dickson (1991) and Kwame Bediako (1995) had previously pointed to the importance of theologising in one's mother tongue before Ekem published his DTheol. dissertation in an attempt to introduce MTBH as a method of biblical interpretation (Ekem 2009). Kuwornu-Adjaottor proposed nine steps for doing MTBH (Kuwornu-Adjaottor 2015, pp. 17–18). Amevenku and Boaheng have suggested five steps which have been used in this paper. The steps are identification of the problem passage, exegesis of the passage, a comparative mother-tongue study of the passage, exploring a culturally appropriate rendition of the passage and a proposal for a new mother-tongue translation of the passage (Amevenku and Boaheng 2022, pp. 90–96).

It is clear from the steps involved in MTBH that the method embarks upon a project not only to understand and interpret a given Bible passage in a culturally valid mother-tongue way, but also seeks to contribute to the improvement of existing mother-tongue translations in the particular language, in this case Eυe. The idea of using MTBH to improve existing vernacular translations is at the same time a decolonial project and a revision project. It is decolonial in the sense that it examines the earliest mother-tongue version of the Bible in the language under consideration before turning to subsequent revisions or new translations into the same receptor language. The earliest translations in West and Southern Africa, at least, are missionary, colonial Bibles.

Mother-tongue biblical interpretation, like intercultural exegesis or decolonial Bible reading, accommodates any valid exegetical approach that takes seriously both the context of the Bible text and the context of the receptor community. Therefore, in doing MTBH, we shall engage to various degrees, the world behind, the world inside and the world in front of the text to bring out the issues of concern in Eph 2:11–22, with emphasis on 2:11 as a source of ethnic disharmony in Ghana.

The entire pericope (2:11–22) gives a basis for the promotion of ethnic harmony in the Ghanaian context, which culminates in a strong bond of unity in 2:13 for all those who accept to be Christ followers. Before getting to 2:13, however, we shall examine how the four Eυe versions of 2:11 illustrate the main concerns of MTBH. The 2:11 rendering of *Biblia* (1931), *Agbenya La* (2006), *Biblia* (2010) and *Agbenya La* (2020) are listed below in succession for analysis in the next section.

The Greek of Eph. 2:11 reads: διὸ μνημονεύετε ὅτι ποτὲ ὑμεῖς τὰ ἔθνη

ἐν σαρκί, οἱ λεγόμενοι ἀκροβυστία ὑπὸ τῆς λεγομένης περιτομῆς ἐν σαρκὶ

χειροποιήτου, τὰ ἔθνη ('Gentiles'); ἐν σαρκί (lit. 'in flesh') repeated in the text, forms a chiastic pattern that many scholars recognise and admit (cf Hiel 2007; Merida 2014) οἱ λεγόμενοι and τῆς λεγομένης also form a chiastic pattern.

The passage may be translated: Therefore, remember that formerly, you Gentiles in (the) flesh who are called 'uncircumcision' by those who are called 'the circumcision' which is made in (the) flesh by human hands. . .

### 3.1. The Eʋe Translation of Ephesians 2:11

Ephesians 2 presents a concept of unity based on faith in Christ Jesus that enables us to appreciate the role of divine intervention in the attainment of cosmic harmony. Cosmic harmony can hardly be separated from ethnic harmony. In Verse 11, there is a reference to Jewish attitudes towards non-Jews, including how they tend to label them as ethnically 'other'. τὰ ἔθνη (the Gentiles) ἐν σαρκί (in the flesh) are characterised/labelled pejoratively as ἀκροβυστία (uncircumcision). How did it come to mean 'idol worshippers' in the 1919 Missionary (Colonial) Eʋe Bible and its 1931 revision? Presented below are four Eʋe versions of Ephesians 2:11 for our analysis.

### 3.2. Ephesians 2:11 According to the Missionary Eʋe Bible (Biblia 1919, 1931)

Eyaŋuti la miɖo ŋku edzi bena, tsã la mi amesiwo nye trɔ̃subɔlawo le ŋutila me, amesiwo woyɔna be aʋamatsomatso le amesiwo woyɔna be aʋatsotso, si wotsɔ asi wɔ le ŋutilã me la gbɔ la,. . . (Because of this remember, that in the past, those of you who were idol worshippers in the flesh, who were called the uncircumcised by those who were called circumcised by human hands. . .)

### 3.3. Ephesians 2:11 According to Agbenya La (2006)

Eyaŋuti la miɖo ŋku edzi bena, tsã la mi amesiwo nye trɔ̃subɔlawo le ŋutila me, amesiwo woyɔna be aʋamatsomatso le amesiwo woyɔna be aʋatsotso, si wotsɔ asi wɔ le ŋutilã me la gbɔ la (Do not ever forget that in the past, you were idol worshippers and Jews called you ungodly and impure. Yet though they carefully observe the rites and rituals of those who fear God, their hearts were not pure, even though they had been circumcised as a sign that they fear God).

### 3.4. Ephesians 2:11 According to Biblia (2010)

Miɖo ŋku edzi be menye Yuda vidzidziwo mienye o. Yudatɔwo yɔa mi be bolobolotɔwo. Eʋo la, aʋatsotso sia la, ŋutilãmenu ko wònye, eye amegbetɔ koe tsɔ asi wɔe (Remember that you are not Jews by birth. Jews refer to you as the uncircumcised. Yet, this circumcision is merely a physical thing and it was carried out by human hands).

### 3.5. Ephesians 2:11 According to Agbenya La (2020)

Eya ta miɖo ŋku edzi be tsã la, mi ame siwo menye Yudatɔwo tso dzidzime o, eye ame siwo yɔa wo ɖokuiwo be 'aʋatsotsotɔwo' (nu si amewo tsɔa asi wɔnɛ) la yɔa mi be 'aʋamatsomatsotɔwo' la, (Therefore, remember that in the past, those of you who are not Jews by birth and those who call themselves the circumcised (something done by human hands), called you 'uncircumcised'. . .)

## 4. The Eʋe Version Compared with Two Twi (Ghanaian) Versions

In Ghana, and for that matter among Eʋe Christians, the Bible is held in high esteem as 'the word of God'. For this reason, any misleading and inadequate translation in the mother tongue, such as rendering *ta ethnē* as 'idol worshippers', strongly influences theologising in that specific mother tongue. This implies that because of their reverence for the Bible as God's word, many, if not most, Eʋe Christians will accept the colonial Bible's verdict that they are 'idol worshippers' simply because they are not Jews. This problem is a source of ethnic disharmony because, as the argument often goes, if some Africans themselves describe Africans as idol worshippers, how can any African dare to challenge that view? They must all be idol worshippers. This is not an acceptable argument, regardless.

While it is true that evidence of inter-tribal and inter-ethnic conflicts existed and were probably rampant in pre-colonial Africa, as some Africans themselves have acknowledged,

that does not mean that colonialism did not contribute to ethnic disharmony in Africa. In fact, colonialism, with its associated imperialism and neo-colonialism in varied forms, still contributes to ethnic disharmony in Africa and especially in West Africa today. It has been argued that by the turn of the 20th century, the map of Africa looked somewhat like a huge jigsaw puzzle as a result of the partition of Africa when European power brokers sat in their offices in a foreign land to draw new boundary lines for African countries they claimed ownership over (Western Colonialism, n.d., Encyclopaedia Britannica, online, 6 November 2023 at 8:04 a.m.). In Africa, Germany announced its claim to several territories in south West Africa, including German Togoland, where the Eʋe were a major ethnic group. As it turned out, German superpower ambition coupled with Britain's and the European Allied Forces' ambition to succeed in WW1 distorted the ethnic Togo Eʋe.

Four hundred years of the slave trade and one hundred years of colonialism resulted in what Englebert Mveng (1930–1995) has called the 'anthropological pauperisation' (Kibangou 2022) of Africa, which partly meant the mutilation of African identities and ethnicities. Similarly, for instance, 'Through its colonial policy of assimilation, France effectively denied the right of Africans to be different and sought to abolish their unique African identity' (Iheanacho 2021, p. 6). An obnoxious policy such as this cannot be explained away by the simple argument that a glorious, pre-colonial Africa devoid of ethnic disharmony was a myth. The fact that a people, in the past, had difficulty dealing with their inter-tribal and inter-ethnic conflicts gives no right to other people to dehumanise and devalue them. The so-called Francophone Eʋe in Togoland was and continues to be affected by this policy.

The 2012 version of the Akuapem Twi (Ghanaian language) Bible renders *ta ethnē* as *amanaman mufo* (foreign nations/people). The 1964 version of the Asante Twi Bible renders it *amanaman mufoɔ*, the Asante Twi version of 'foreign nations'. These Twi versions preserve the Greek idea, but 'idol worshippers' does not. On the contrary, the Eʋe Missionary Bible and its revisions prior to 2010 introduce a new, foreign element that is capable of distorting Eʋe further if believed.

## 5. The Problem of Ethnic Disharmony in Ghana

Mr Kennedy Agyapong, the Ghanaian member of parliament for Assin North in 2023, is reported in 2012 to have said at Oman FM, his radio station, that, 'Today I declare war in this country, Gbevlo-Lartey and his people, IGP should know this. Voltarians in the Ashanti region will not be spared. If anyone touches you, butcher him with a cutlass...' Gbevlo-Lartey was the coordinator of national security at that time, and Eʋe is the predominant tribe of the then Volta region of Ghana. By 'Voltarians', therefore, most people understood Mr Agyapong to be referring to the Eʋe. A strong backlash followed the unguarded comments. Mr Agyapong has never regretted this comment. On the contrary, he continues to spew out, in deadly anger, insults and vituperations about people he dislikes for one reason or the other. No one seems to know how to deal with Kennedy Agyapong and the tension and insecurity he fuels by his utterances.

On 11 June 2023, Kennedy Agyapong addressed a rally at Hohoe in the Volta region (predominantly Eʋe), where he sought to debunk allegations made against him that he is anti-Eʋe and does not deserve Eʋe support in his flag-bearer-hopeful campaign (Agyapong 2023, www.ghanaweb.com, accessed on 23 July 2023). Agyapong claimed that because he had given the Eʋe people 'two smart girls', he deserved their praise and support.

A popular charismatic church leader and founder of Perez Chapel International (formerly the Word Miracle Church), Archbishop Charles Agyinasare, received strong pushback from Nogokpo, an Eʋe community, when he said in one of his sermons that 'Nogokpo is the demonic centre of the Volta Region'. Though the drama has died down quite a bit, at the time of writing this paper, it is likely that the people will revisit the comment another time. The Archbishop drew both strong support and scathing criticism from different people at different times since his comments in June 2023. He continues to publish defiant rhetoric, signalling no desire to seek reconciliation with the community he offended with his unguarded comments.

The problem of labelling people in Ghana by where they come from is a common one. It plays out in many other forms, including conversations in the religious arena. For example, the Evangelical Presbyterian (EP) Church of Ghana is one of the earliest churches to be planted in Ghana. It is one of two Presbyterian churches in Ghana. It started among the Eʋe in 1847 (Amevenku 2019) and has since grown to become a major denomination in Ghana, with its overwhelming membership coming from the Volta region (Now Volta and Oti regions). Though the church has contributed significantly to the development of Ghana, in terms of education, health care provision, skill training and human resource development, the fact of its predominantly uni-ethic character, in the view of some Ghanaian people, makes the church an object of scorn. For instance, most people in the Ashanti region (Ashanti Twi-speaking) refer to the EP Church simply as 'Ayigbe Presby', by which they mean the Eʋe Presbyterian Church. To be sure, the church used to be called 'Eʋe Presbyterian Church', but realising that its mission had gone far beyond Eʋeland, it changed its name to the Evangelical Presbyterian Church (Amevenku 2019). Those who love to label the EP Church, however, cannot be bothered by the change. This attitude of labelling causes serious ethnic disharmony in Ghana. It is a major challenge to mission and evangelisation as well.

## 6. Labelling as a Source of Ethnic Disharmony in Ghana

Some Ghanaian people label other Ghanaian persons simply because those they label come from different ethnic groups. Many Ghanaian people call the Eʋe Ayigbe[1], as noted above. There is a popular Ghanaian musician who is known simply as 'Ayigbe Edem'. Edem is an Eʋe name, meaning 'He (God) has delivered me'. The musician probably did not put 'Ayigbe Edem' on himself. Indeed, there is a community in Nsawan, a town near Accra, the capital of Ghana, which is called 'Ayigbe Town' because many Eʋe people live there. In Kumasi, the second city of Ghana, there is a place known as 'Aŋlɔgã Junction' because, as far as the people of Kumasi are concerned, *Aŋlɔ* is their label for Eʋe (though, as mentioned above, they also label the EP Church as 'Ayigbe Presby'). The original *Aŋlɔgã* is a prominent town in the Ketu south district of the Volta Region.

For their part, many Eʋe people refer to the Akan person as *Bluvi*, though the Akans do not call themselves by that label. Some Akans call a person of northern Ghana origin, *Té-ni*. The people of the regions of northern Ghana also have a derogatory label for all the Ghanaian people who come from the southern regions. How can these ethnic-related tensions be managed to prevent their degeneration into full-scale conflicts? It is argued, in this paper, that the Christian gospel offers a solution through mother-tongue theologising based on careful mother-tongue hermeneutics faithfully applied.

## 7. The Search for a Culturally Appropriate Language for Theologising

Language is a crucial identity marker. Language is part of culture, and a people's cultural identity is their pride. Whoever takes a person's language from them, or deprives someone of their culture, destroys the victim's cultural identity. The significance of cultural identity is not lost on the cultures of the Bible. Jewish culture, in particular, is a source of great pride in the Hebrew Bible. The desire to preserve Jewish cultural identity as a function of Jewish loyalty to the Jewish God made prominent leaders such as Nehemiah to commit what some 21st-century human rights advocates could describe as 'ethnic cleansing' (cf Ezra 10:3ff; Neh. 13:23ff). Nehemiah attacked some male Jews of his day, pulling their hair and beating them up, because their children could only speak a foreign language (Neh 13:23–30) but not their 'father tongue'. Ezra acted similarly to purge the community of foreign wives and children (see 10:1–3, 5).

What about their mother tongue? If the 'father tongue' is so important to the Jews and perhaps their God as to lead to the drastic reforms described above, should the mother tongue not be so important as to warrant a method of biblical interpretation for vulnerable and socially and culturally excluded people such as the Eʋe people of West Africa? Mother-tongue hermeneutics, then, in my view, is a decolonial hermeneutic, and

thankfully, the adherents of decolonised Bible reading say the approach admits of any valid method of interpretation, which shares the goals and ideals of decolonisation of biblical interpretation.[2]

*The Greek Text of Ephesians 2:11 (the UBS Greek New Testament) and Its Eʋe Rendering*

Διὸ μνημονεύετε ὅτι ποτὲ ὑμεῖς τὰ ἔθνη ἐν σαρκί, οἱ λεγόμενοι ἀκροβυστία ὑπὸ τῆς λεγομένης περιτομῆς ἐν σαρκὶ χειροποιήτου.

## 8. Analysis of the Eʋe Versions

It has been established that the word Trɔ̃subɔlawo (idol worshippers) distorts the meaning of the Greek text of Eph. 2:11 for the Eʋe people. The missionaries started it in 1919. The revised version of 1929 maintained the distortion, and not even the 1931 revision or the 1996 and 2006 reprints bothered to correct the colonised translation's reference to those who are not Jews as 'idol worshippers'. It is understandable that not only the initial missionary colonised Eʋe Bible but its subsequent revisions and the indigenous translations were all funded by westerners. That does not mean that the translators could not have brought their concerns to bear on the translated text. The 2010 and 2020 versions of the Eʋe Bible avoided the earlier translations of τὰ ἔθνη as trɔ̃subɔlawo.

The Bible Society of Ghana, a division of the United Bible Societies, produced the 2010 version. Biblica Ghana, a division of the International Bible Society, produced the 2020 version. Elsewhere, in those two versions, however, τὰ ἔθνη is again translated trɔ̃subɔlawo for no apparent reason. Perhaps only the notes of the translators and the consultant might help us understand why the translators made that choice. In any event, the brief analysis of Eph. 2:11 already reveals how MTBH exposes the colonised nature of Bible translation in Africa and underscores why a decolonial reading of the text in the mother tongue is valid. A decolonised reading of the text will reject the reality of name calling (Larkin 2009, p. 37) underscored by the phrase οἱ λεγόμενοι Like Kuwornu-Adjaottor's nine-step method, Amevenku's and Boaheng's five-step method climaxes in the proposal of a new translation, which is offered as follows: Eya ta miɖo ŋku edzi be tsã la, mienye Yudatɔwo tso dzidzime o, eye Yudatɔ, siwo yɔa wo ɖokuiwo be 'ametsoaʋawo' (nusi wotsɔ asi wɔ) la yɔa mi be 'amematsoaʋawo' (Therefore remember that in the past, you who are not Jews by birth, the Jews, who call themselves 'the circumcised' (something done with human hands), called 'uncircumcised'). This proposed mother-tongue translation is an example of how to reject imperialism and colonisation in Bible translation and theologising. It is a way to avoid 'hermeneutical hegemony and ideological stranglehold of Eurocentric Biblical Interpretation' (Drapper 2015).

## 9. Implications for Christian Unity and Ethnic Harmony

John Stott observes that Eph 2:11–22 presents 'a portrait of alienated humanity' (Stott 2007, p. 94). Here, it is sin that alienated sinful Gentiles from the commonwealth of Israel. At the same time, disobedience and rebellion alienated national Jews from their God. According to Toby Eisenberg, 'Ephesians *is* an ecumenical document' (Eisenberg 2019, p. 17). Central to ecumenism is the concept of unity in diversity. Frederick F Bruce notes that being reminded constantly of what you were apart from God's grace enables you to appreciate the grace of God and overcome the temptation of thinking too highly of yourself (Bruce 2012, p. 155). For Martin Kitchen, Paul mediates a religious tradition in Ephesians 2 that advertises the church's self-understanding that the church 'on earth' exudes the end of ethnic hostility between Jews and Gentiles (Kitchen 2002, p. 29).

One reason for ethnic labelling is the false sense of importance that easily overcomes a person who is blinded to his or her own limitations. It leads to complacency and an exaggerated self-importance (MacArthur 2015). For instance, the Jews of Paul's day boasted in their circumcision as a 'mark of covenantal privilege, social standing, and spiritual purity' (Chapel 2009, p. 89), and for many of us today, others tend to exclude us because of our accent, parentage, origin, or perhaps height. The Jews were 'basing their identity only' on

something done in the flesh with human hands: 'our humanity inevitably shows through in our pettiness, our quarrels, our lack of forgiveness, our envy, our lusts, our gossip, and our laziness' (Chapel 2009, pp. 89–92). This exaggerated self-importance is often communicated in both word and deed. However, the person who belongs to Christ, who has a sense of his or her own sinfulness, values the grace of God that brought him or her to salvation in Christ. If, however, a person fails to recognise the value of the gift of salvation they possess, they despise who they are, as well as the Lord who saved them. As Bruce has noted, 'No iron curtain, color bar, class distinction or national frontier of today is more absolute than the division of Jew and Gentile was in antiquity' (Bruce 2012, p. 161). The apostolic gospel overcame this deep estrangement and enabled Jew and Gentile to live in unity as God's children, because 'Those who enter peace with God have peace with one another' (Bruce 2012, p. 161).

Paul uses building imagery to develop the theme of unity in this epistle. He refers to a building located in a city (or town) occupied by families (2:19–22) (see also Eph 3:5; 4:11; 1 Cor 3:11; 1 Cor 12:28). Commenting on Verse 2:21, Bruce further takes the view that since Paul uses the biological language of the building growing, he must have been conceiving of church as a living organism and the growth envisaged as building this living organism rather than a 'growing building' as some translations portray (Bruce 2012, p. 177). A reference to alienation in 2:12 imports political language relating to citizenship to illustrate the nature of the hostility between Jews and Gentiles (Kobelski 2000, p. 887). Before the onset of the bestowal of the grace of Christ to be appropriated by faith, the Gentiles were alienated from the commonwealth of Israel, excluded from the community of God's people. Now that the Gentiles have been brought in and unified with the Jews in one humanity, the redeemed people of God, all hostility must cease.

## 10. Implications for Ethnic Harmony in Ghana

What people believe about themselves shapes their identity, and identity is formed in many ways through socialisation at home, in the clan, the family and the town or nation, among other ways. Socialisation is one main source by which the interplay between tradition and social change unfolds (Nukunya 2003). This is often reflected in the culture of a people, including their religion and how they theologise. In Ghana and among the Eʋe, the Bible is the main source of Christian theologising, even though Christian tradition, reason and experience also contribute to theological discourse. Therefore, the way the Bible is translated greatly influences what the people believe about God and themselves. This belief also influences what they do because belief is action-oriented. By telling the Eʋe people they are 'idol worshippers' through the powerful medium of Bible translation, the missionary, colonial Bible translators succeeded without probably intending to, in contributing to the identity distortion and cultural erosion of the Eʋe people.

Coupled with the big negative influences of the twin evils of slavery and colonialism for 500 years, with their associated arbitrary distortion of boundary lines in Africa, 'with little or no attention to ethnic unity, regional economic ties, tribal migratory patterns, or even natural boundaries' (Western Colonialism, n.d., Encyclopaedia Britannica, online, 6 November 2023), the colonised Eʋe Bible has contributed to ethnic disharmony in Ghana. The Eʋe people, a large, West African ethnic group of about 6 million people, through the slave trade, colonialism and the colonial Christian missions, have been scattered politically across West Africa. They are found in Ghana, Togo and Benin, as well as southern Nigeria, to a lesser extent. With this development, fuelled further by artificial political colonial francophone and anglophone borders, the ethnic harmony of the Eʋe has been destroyed forever. In Ghana, some Ghanaian people are quick to label the Eʋe people as 'foreigners' even though the entity called 'Ghana' was non-existent before 1957. All of this causes ethnic tension and disharmony in Ghana. This paper has argued that MTBH offers a potential for the restoration of ethnic pride and identity for the Eʋe people, just like any other people who are desirous to do more for themselves to ensure that they live in dignity devoid of ethnic dehumanisation, devaluation and labelling by other people.

## 11. Conclusions

Ethnic disharmony is a problem in Ghana. Rather than use its rich ethnic and cultural diversity to its advantage, Ghana has struggled to deal with the challenge of multiple ethnicities, peoples, languages, states and cultures. Though about 72% of the Ghanaian people profess to be Christian, Ghanaian Christians have not succeeded in bringing beautiful Christian virtues such as unity in diversity to bear on the nation and its diverse people and ethnicities. On the contrary, some Ghanaian politicians have been exploiting the fact of Ghana's ethnic diversity to their own selfish, personal 'advantage' without realising the deep-seated problems they create for the nation.

In Judaism, disobedience and rebellion alienated the Jews, God's chosen people, from their God. Sin alienated humanity from God as well, as far as Christianity is concerned. It is sin that alienated sinful Gentiles from the God of Jesus. Jesus came to heal that alienation and to reunite both Jews and Gentiles into one humanity. For this reason, those who have believed in Jesus ought to emulate his life as a unifier. Unfortunately, Ghanaian Christians, like other Christians around the world, have not lived up to this glorious Christian unity, attained through the sacrifice of peace, leading to reconciliation in Christ (Eph. 2:11). Instead, Ghanaian Christians have allowed ethnic bigotry and ethnocentrism to lead to ethnic disharmony, among them, which is responsible for sectarian politics, political tensions and lack of development in Ghana. Colonial, missionary Bible translation, rather than affirming the reality of diversities and ethnicities of different peoples, including the Eʋe, unfortunately contributed to the identity distortion and labelling that continues to fuel disharmony. Using MTBH to read Eph 2:11, this paper has suggested one approach to addressing ethnic labelling among Ghanaian Christians and promoting ethnic harmony, Christian unity, collegiality and ecumenism in Ghana.

**Funding:** This study received no external funding.

**Conflicts of Interest:** The author declares no conflict of interest.

## Notes

[1] Margaret Sagoe (2020) (recalls a Twitter comment attributed to @Carl_Enam) "This is Why Ewes are Sometimes Called 'Ayigbe'", 21 June 2020. In Ghana, many Akans refer to the Eʋe as *Ayigbe-fuɔ* and many Ga people call them *Ayigbe-tsɛmɛin*. These labels offend some Eʋe people but others do not mind. The Eʋes and the Gas have historical ties. They migrated to their present locations around the same period. When the Eʋes arrived at Ŋɔtsie in present-day Togo, the Gas soon joined them, and they lived together. When the Ga population expanded, they left. The Gas moved to Accra in present-day Ghana. Others remained in present-day Togo and were led by the Crown Prince, Ayi. The King led those who moved. Ayi had custody of some of the royal items. When the King died, the Accra Gas sent a delegation to Ayi to urge him to join his people in Accra and take his rightful place as King. But through intermarriage between the two ethnic groups, Ayi had been installed King. Ayi decided not to go to Accra and sent a message saying, *yaa kɛɛ amɛ akɛ,* (go tell them, i.e., Ga) *'Ayi gbɛ'* (Ayi says, 'No', i.e., Eʋe). The delegates returned and mispronouced *Ayi gbɛ* as Ayigbe. When the Gas celebrate their annual Homowo festival, they some of the rites in Togo. Source: Pemtsikata.wordpress.com, accessed on 29 November 2023.

[2] Professor Itumeleng Mothoagau shared this view at Legon, Accra, in June 2023 at the Sacred Text Conference organised by the Department for the Study of Religions.

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
