# Peer review of "Mother-Tongue Biblical Hermeneutics and the Pursuit of Ethnic Harmony in Ghana"

_religions, doi:10.3390/rel14121491_

Round 1
Reviewer 1 Report
Comments and Suggestions for Authors
The author proposes to apply mother tongue biblical hermeneutics (MTBH) to the interpretation of Eph 2:11. But the author becomes distracted by an account of ethnic disharmony in Ghana, commentary on Neh 13:23-30 and Ezra 10:1-3, a general overview of the Letter to the Ephesians, a historical account and description of MTBH, an introduction to the Ewe people of West Africa, an account of colonizing missionary activity among the Ewe, an account of political conflict in Ghana, a critique of European influenced Ewe translations of Eph 2:11, and then finally, but briefly, the author proposes a decolonized Ewe translation of Eph 2:11. But there is no in-depth exegetical treatment of Eph 2 or any part thereof using MTBH. The author covers too many topics in this essay, and therefore, does not provide the depth of analysis needed for an exegetical and hermeneutical project. The author should have started with an exegesis of Eph 2:11 and context using MTBH and then apply the text to the cultural and political situation of Ghana in the final few paragraphs of the paper.
Comments on the Quality of English LanguageThis essay shows good command of the English language. However, there are a few minor spelling errors. The misspelled words are "published" (line 136), "when" (line 295), and "poses" (line 419). Also, there is repetition of the word "is" (line 247).
Author Response
Dear Reviewer,
Thank you for your review comments which I found pertinent and helpful. I have addressed your concerns as thoroughly as possible within the time constraints. Please see the attachment.
Reviewer 2 Report
Comments and Suggestions for Authors
The article's argument is that Ghana, an amalgamation of different states, ethnic groups, people and languages, represent different cultures and concerns. Some Ghanaian politicians exploit this diversity to their advantage and the nation's detriment, resulting in ethnic disharmony. This is supported by missionary Christianity, who historically reduced all who did not subscribe to Christianity as social nonentities. Conversion thus led to the erosion of cultural identity. To address this challenge, it suggests utilising Mother-Tongue Biblical Hermeneutics (MTBH) to improve existing vernacular, missionary and colonial Bible translations. It is a decolonial project in the sense that it examines the earliest mother-tongue version of the Bible in the language involved before turning to subsequent revisions or new translations into the same receptor language. It applies the method to Ephesians 2 to argue for Christianity as bringing harmony between ethnic groups.
Both the introduction and conclusion are extremely concise and do not provide the necessary information to present the argumentation in the work.
The exegesis of Ephesians 2 rests nearly solely on the work of Kobelski 2000.
The article is mainly an argument the author(s) developed without referring to relevant resources. For that reason, the Bibliography is very short.
It is suggested that the argument needs to be revised to reference it more adequately in terms of existing relevant literature and that the abstract and conclusion should be more informative of the argument in the paper.
Comments on the Quality of English LanguageUse of language is relatively good.
Author Response
Dear Reviewer,
Thank you for your comments and suggestions. I have addressed them as much as I can within the constraints of time. Please see the attachment.
Reviewer 3 Report
Comments and Suggestions for Authors
1. The article attempts to highlight how Mother Tongue Biblical Hermeneutics may promote ethnic harmony in Ghana by addressing the imperial and colonial prejudices in the Ewe Bible. The article centres on the translation of the phrase in Ephesians τὰ ἔθνη as Trɔ̃subɔlawo (idol worshippers).
2. I have indicated several places within the paper that need further clarification.
3. While the paper makes very interesting reading, I find that it is based on problematic assumption of a glorious harmonious pre-colonial Africa that was destroyed by Western colonialists. Yet, there are many historical accounts by Africans themselves that show pre-colonial communities were deeply divided and characterised by tribal superiorities that often led to serious inter-tribal conflicts. While it is fair to criticise western missionaries for poor and imperialistic translations of the Ewe Bible, I find it factually problematic to attribute ethnic disharmony among the people of Ghana to Western missionaries’ hermeneutics as suggested by the article.
4. Furthermore the link between the translation of τὰ ἔθνη as Trɔ̃subɔlawo (idol worshippers) and ethnic disharmony in Ghana is not very clear. I wish this to be clarified.
5. I would like to see the article published provided the above concerns are addressed.

Author Response
Dear Reviewer,
Thank you for your comments and suggestions which I find pertinent and helpful. I have addressed them as much as I can within the constraints of time. Please see the attachment.
Reviewer 4 Report
Comments and Suggestions for Authors
From the editorial viewpoint I suggest some revisions of English language in some parts of the article. Secondly, some acronymous shlould be better explained in order to understand the main points of the presentation. Some editing revision is required as portions of the paper are not righlty place (viz. some appear to be like a quotation and they are not and/or spacing is not consistent in many points)
From the content perspective, I suggest a more accurate connection and consistency between the hypothesis and the Scriptural analysis.
Conclusion should be better substatiated referring to the previous part of the study.
Comments on the Quality of English Language
From the editorial viewpoint I suggest some revisions of English language in some parts of the article. Secondly, some acronymous shlould be better explained in order to understand the main points of the presentation. Some editing revision is required as portions of the paper are not righlty place (viz. some appear to be like a quotation and they are not and/or spacing is not consistent in many points)
From the content perspective, I suggest a more accurate connection and consistency between the hypothesis and the Scriptural analysis.
Conclusions should be better substatiated referring to the previous part of the study.
Author Response
Dear Reviewer,
Thank you for your comments and suggestions, which I find useful and pertinent. I have addressed them as far as I can within the constraints of time. Please see the attachment.
Round 2
Reviewer 1 Report
Comments and Suggestions for Authors
The revised manuscript is a great improvement. However, I am confused by several problems that remain: Lines 279-293 is a repetition of lines 35-48. Lines 305-344 is a repetition of lines 49-74. It seems that lines 294-304 should follow line 48. It seems that lines 364-390 should follow line 239. There is a lacuna (gap) between lines 66-67. I do not know whether this apparent jumbling of the text occurred in transmission or not. In any case, these are issues that will need to be resolved by either the author, editor, or typesetter. When these issues are resolved, the article will be ready for publication.
Author Response
Dear Reviewer,
Many thanks for your review comments which I found helpful.
Please see the attached.

Reviewer 2 Report
Comments and Suggestions for Authors
The article has been revised carefully and can be published except for the exegetical commentary on key texts. Exegesis does not consist of the following but requires the skill to distract information from the text that attributes to the theme under discussion:
'In the Greek, Paul writes in Eph. 2:11: διὸ μνημονεύετε ὅτι ποτὲ ὑμεῖς τὰ ἔθνη ἐν σαρκί, οἱ λεγόμενοι ἀκροβυστία ὑπὸ τῆς λεγομένης περιτομῆς ἐν σαρκὶ χειροποιήτου, The morphology of the passage follows: διὸ is a conjunction meaning ‘therefore.’ μνημονεύετε is the present imperative active second person plural verb meaning ‘remember’; ὅτι is the conjunction ‘that’; ποτὲ is an adverb qualifying the time of reference and it means ‘formerly’; ὑμεῖς is the second person personal plural nominative possessive pronoun ‘you’; τὰ ἔθνη is the nominative neuter plural noun meaning ‘the Gentiles’; ἐν σαρκί (lit. ‘in flesh’) is a noun feminine singular dative meaning ‘in [the] flesh.’ Repeated in the text, it forms a chiastic pattern that many scholars recognise and admit (cf Hiel 2007; Merida 2014) οἱ λεγόμενοι ἀκροβυστία; λεγόμενοι is either middle or passive and can be translated ‘called’ or ‘being called’. Thus, οἱ λεγόμενοι ἀκροβυστία means ‘the ones called uncircumcision or ‘the ones being called uncircumcision.’ ὑπὸ is a preposition meaning ‘by’τῆς λεγομένης can be translated ‘that being (passive voice) called’ or ‘those (who are) called’(middle voice). οἱ λεγόμενοι and τῆς λεγομένης also form a chiastic pattern. Περιτομῆς is feminine genitive singular noun meaning of circumcision. ἐν σαρκὶ has been commented on above. Χειροποιήτου is a genetive singular adjective of manner describing how the circumcision mentioned was made (by human hands)'
Comments on the Quality of English LanguageNone
Author Response
Dear Reviewer,
Many thanks for your review comments. They are helpful.
Please see the attached.
